# LONGCITE: ENABLING LLMS TO GENERATE FINE-GRAINED CITATIONS IN LONG-CONTEXT QA

## ABSTRACT

Though current long-context large language models (LLMs) have demonstrated impressive capacities in answering user questions based on extensive text, the lack of citations in their responses makes user verification difficult, leading to concerns about their trustworthiness due to their potential hallucinations. In this work, we aim to enable long-context LLMs to generate responses with fine-grained sentence-level citations, improving their faithfulness and verifiability. We first introduce LongBench-Cite, an automated benchmark for assessing current LLMs' performance in Long-Context Question Answering with Citations (LQAC), revealing considerable room for improvement. To this end, we propose CoF (Coarse to Fine), a novel pipeline that utilizes off-the-shelf LLMs to automatically generate long-context QA instances with precise sentence-level citations, and leverage this pipeline to construct LongCite-45k, a large-scale SFT dataset for LQAC. Finally, we train LongCite-8B and LongCite-9B using the LongCite-45k dataset, successfully enabling their generation of accurate responses and fine-grained sentence-level citations in a single output. The evaluation results on LongBench-Cite show that our trained models achieve state-of-the-art citation quality, surpassing advanced proprietary models including GPT-4o. We also discover that SFT with citation information can further improve the correctness of model responses compared to standard long-context SFT.

## 1 INTRODUCTION

Recent years have witnessed significant advancement in long-context large language models (LLMs), enabling them to address various user questions, such as information extraction and summarization, based on lengthy texts that surpass 100,000 tokens (Anthropic, 2024b; Zeng et al., 2024; Reid et al., 2024). Despite their remarkable capacities, current long-context LLMs typically do not provide citations to specific context snippets to support the statements they generated, making it challenging for users to verify model outputs given the substantial context lengths. This significantly impacts the reliability and trustworthiness of long-context LLMs, especially considering that they still struggle with hallucinations (Huang et al., 2023) and are prone to generate unfaithful content.

On the other hand, recent works in search engines and open-domain QA have allowed LLMs to generate responses with in-line citations through retrieval-based generation (RAG) or post-hoc methods (Nakano et al., 2021; Gao et al., 2023a;b; Menick et al., 2022). Nevertheless, these approaches still expose notable limitations in long-context scenarios: RAG often leads to compromised answer quality due to incomplete context information, while post-hoc methods prolong the user waiting time due to more complicated pipeline. In addition, their generated citations typically refer to entire web pages (Nakano et al., 2021) or coarsely chunked snippets (Gao et al., 2023b), thereby requiring users to further pinpoint the specific supporting evidence for the final verification.

To overcome the above limitations, this work explores directly employing long-context LLMs to generate accurate responses with fine-grained sentence-level in-line citations. We first propose **LongBench-Cite**, an automatic benchmark, to evaluate LLMs' performance on the task of **long-context question answering with citations (LQAC)**, and find that current LLMs obtain unsatisfactory results (Sec. 2). Specifically, we find that many citations produced by current LLMs are either irrelevant, cannot fully support the response, or have a coarse granularity. Meanwhile, we observe

that generating citations on the fly via in-context learning generally results in responses with lower correctness compared to vanilla long-context QA.

To further enhance the inherent capacity of LLMs for generating fine-grained citations from lengthy contexts, it is essential to construct a high-quality SFT dataset. To this end, we introduce **CoF** (abbr. for "**Co**arse to **F**ine"), a novel pipeline that utilizes off-the-shelf LLMs to automatically construct long-context QA instances with precise sentence-level citations (Sec. 3). CoF comprises four stages: (1) Starting with a long text material, CoF first invokes the LLM to produce a query and its associated answer through Self-Instruct (Wang et al., 2023). (2) Next, CoF uses the answer to retrieve several chunks (each has a fixed length of 128 tokens [1]) from the context, which are then fed into the LLM to incorporate coarse-grained chunk-level citations into the answer. (3) The LLM subsequently identifies relevant sentences from each cited chunk to produce fine-grained citations. (4) As a final step, instances with an insufficient number of citations are discarded. Our experiments validate the superiority of CoF over other LQAC strategies in terms of answer correctness and citation quality. With CoF, we construct **LongCite-45k**, a large-scale SFT dataset that consists of 44,600 high-quality LQAC instances with contexts up to 128,000 tokens.

Finally, we utilize LongCite-45k to fine-tune GLM-4-9B (Zeng et al., 2024) and Llama3.1-8B (Vavekanand & Sam, 2024), two latest open-source long-context models (Sec. 4). The enhanced models, namely **LongCite-9B** and **LongCite-8B**, support a max context window of 128,000 tokens and are capable of generating accurate responses along with precise, fine-grained citations in one pass. Evaluation on LongBench-Cite indicates that our trained models achieve significantly better citation quality compared to even much larger proprietary models. Specifically, our 8B/9B size model outperforms GPT-4o by 6.4%/3.6% in terms of citation F1 score and achieves twice finer granularity. Meanwhile, we observe that SFT with citation information can alleviate hallucinations of LLMs and enable them to utilize context information more uniformly and comprehensively, instead of only focusing on a specific part of the context. This results in a further improvement in response correctness over standard long-context SFT. We also conduct extensive analyses and human evaluation to further verify the effectiveness of our approach.

To summarize, our work makes the following contributions:

1. We introduce LongBench-Cite, an automatic benchmark for the task of LQAC, and reveal the limited performance of current long-context LLMs.

2. We propose CoF, which utilizes off-the-shelf LLMs to automatically construct high-quality long-context QA instances with fine-grained sentence-level citations. Using this method, we construct LongCite-45k, a large-scale SFT dataset for LQAC.

3. We successfully train LongCite-8B and LongCite-9B using LongCite-45k dataset, allowing the generation of accurate responses and fine-grained citations in one pass. Our experiments show that SFT on LQAC data not only enhances the capacity for generating citations from lengthy contexts but also further improves response correctness.

## 2 LONGBENCH-CITE: BENCHMARK LONG-CONTEXT QA WITH CITATIONS

### 2.1 PROBLEM DEFINITION

We formalize the task of **long-context question answering with citations (LQAC)** as follows: given a long context $\mathcal{D}$ and a query $q$, the LLM is required to return a response $\mathcal{A}$, which consists of $n$ statements $s_1, \ldots, s_n$, and each statement $s_i$ cites a list of snippets $\mathcal{C}_i = \{c_{i,1}, c_{i,2}, \ldots\}$ from $\mathcal{D}$. In this work, LLMs need to segment their responses into statements based on semantic integrity by enclosing each statement with two special tokens <statement> and </statement>. As illustrated in Figure 1, we consider two types of citations:

- **Chunk-level citations**, where the context $\mathcal{D}$ is divided into indexed chunks with a fix length of 128 tokens, and each citation $c_{i,j}$ is in the form of $[k]$, referring to the $k$-th chunk;
- **Sentence-level citations**, where $\mathcal{D}$ is divided into indexed sentences, and each $c_{i,j}$ takes the form of $[a\text{-}b]$, referring to the snippet that includes the $a$-th to $b$-th sentences in $\mathcal{D}$.

---

[1]In this work, we uniformly use GLM4-9B's tokenizer to count tokens.

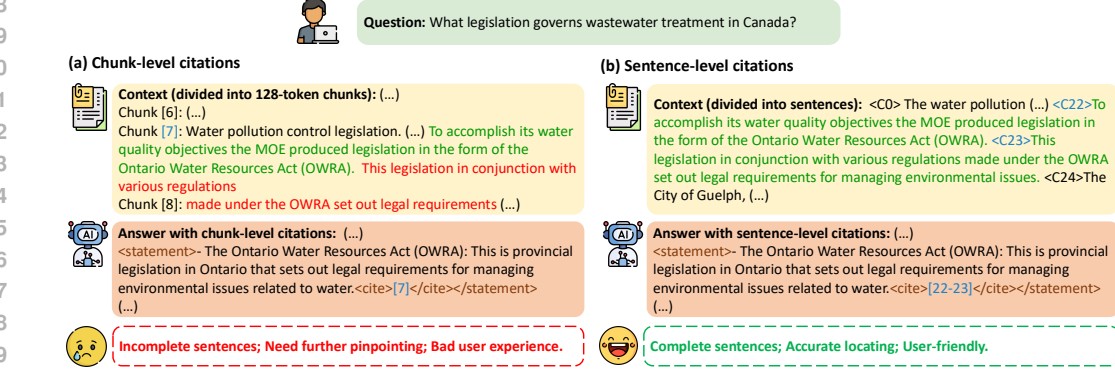

Figure 1: Comparison between chunk-level and sentence-level citations.

| Dataset | Task | Source | Avg Len | Language | #data |
|---|---|---|---|---|---|
| MultiFieldQA-en | Single-Doc QA | Multi-field | 4,559 | English | 150 |
| MultiFieldQA-zh | Single-Doc QA | Multi-field | 6,701 | Chinese | 200 |
| HotpotQA | Multi-Doc QA | Wikipedia | 9,151 | English | 200 |
| Dureader | Multi-Doc QA | Baidu Search | 15,768 | Chinese | 200 |
| GovReport | Summarization | Government Report | 8,734 | English | 200 |
| LongBench-Chat | Multi-task | Real-world Query | 35,571 | English/Chinese | 50 |

Table 1: Data Statistics in LongBench-Cite. 'Source' means the origin of the context. 'Avg Len' denotes the average number of words/characters of contexts in English/Chinese datasets.

Most previous works (Menick et al., 2022; Gao et al., 2023b; Buchmann et al., 2024) for citation generation explore the chunk-level citations. However, the coarse granularity of chunk-level citations requires users to sift through many irrelevant details in the cited content, and the crude segmentation applied for chunk-level citations often results in incomplete cited sentences. Therefore, in this work, we mainly focus on sentence-level citations (Slobodkin et al., 2024; Huang et al., 2024) because they allow for finer-grained citation, ensure semantic integrity better, and are thus more user-friendly.

## 2.2 Data Collection

To evaluate LLMs' performance on LQAC task, we curate a new benchmark **LongBench-Cite** by collecting data from existing bilingual long-context benchmarks LongBench (Bai et al., 2023) and LongBench-Chat (Bai et al., 2024), covering multiple key user-intensive tasks in both English and Chinese. Specifically, LongBench is a comprehensive benchmark with an average length of 7k words (English) and 13k characters (Chinese), and we select two single-doc QA datasets *MultiFieldQA-en/zh* (Bai et al., 2023), two multi-doc QA datasets *HotpotQA* (Yang et al., 2018) and *DuReader* (He et al., 2018), and one summarization dataset *GovReport* (Huang et al., 2021) from it. LongBench-Chat comprises 50 real-world queries with long contexts ranging from 10k to 100k in length, covering various scenarios such as document QA, summarization, and coding, and we adopt all the queries. The detailed data statistics are listed in Table 1. For all datasets, we require LLMs to generate long-form responses with citations.

## 2.3 Automatic Evaluation

LongBench-Cite evaluates models' responses based on the two dimensions:

- **Correctness:** Whether the response is accurate and consistent with the groundtruth.
- **Citation quality:** Whether the response is entirely supported by the cited snippets, no irrelevant snippets are cited, and the cited snippets are fine-grained.

In the following, we introduce automatic metrics for each dimension.

### 2.3.1 EVALUATION OF CORRECTNESS

For the correctness dimension, we adopt the evaluation method of Bai et al. (2024), which is specially designed for long-form responses. Specifically, we first remove citation-relevant tokens from LLM response, then ask GPT-4o to rate the response based on the query and groundtruth answers via few-shot (for LongBench-Chat) or zero-shot prompting (for other datasets). The detailed prompts can be found in Figure 4, 5, and 6. In addition, to investigate whether adding citations will hurt or improve models' long-context QA performance, we propose a new metric **correctness ratio**:

$$CR = C/C_{LQA} \times 100\% \tag{1}$$

Here, C and $C_{LQA}$ respectively denote the correctness in LQAC setting and vanilla long-context QA setting (i.e., simply feeding the concatenated context and query into the LLM to get a response).

### 2.3.2 EVALUATION OF CITATION QUALITY

To evaluate the citation quality, we select **citation F1** calculated using **citation recall** and **citation precision** (Gao et al., 2023b) as the main metric, where the former examines if the model response is fully supported by cited snippets and the later detects irrelevant citations. Compared with Gao et al. (2023b), which uses NLI model TRUE (Honovich et al., 2022) for automatic examination, we further improve the measurement method with GPT-4o to better adapt to long-context QA scenarios. Human evaluation (Sec. 4.3) demonstrates our method has a stronger agreement with human. Besides, we use **citation length** to measure the granularity of citations and avoid trivial results.

**Citation Recall.** We score citation recall (0/0.5/1) for each statement and average over all statements in the model response. Specifically, for each statement $s_i$ that cites at least one snippet (i.e., $\mathcal{C}_i \neq \emptyset$), we concatenate all snippets in $\mathcal{C}_i$ and ask GPT-4o to judge whether the concatenated text fully supports (1 point), partially supports (0.5 point), or does not support (0 point) $s_i$. On the other hand, most LLM responses contain several "functional sentences" such as *"The proposed method has the following advantages:"* and *"In summary, ..."* that do not require citation. Therefore, for each statement $s_i$ that has no citation, we feed $s_i$ along with the query and the whole response into GPT-4o and prompt it to determine if $s_i$ is a starting sentence, transition sentence, or a summary or reasoning based on the previous response content. If so, $s_i$ needs no citation and directly receives a citation recall of 1; otherwise, the recall is 0. The prompts are shown in Figure 7 and 8.

**Citation Precision.** We calculate citation precision for each citation (0/1 for irrelevant/relevant citations) and average over all citations in the response. Here, a cited snippet $c_{i,j}$ is relevant if and only if it entails some key points of the statement $s_i$, i.e., at least partially supports $s_i$. We also employ GPT-4o as the judge using the prompt in Figure 9. In contrast, Gao et al. (2023b) may overlook partially supporting cases due to the limited capacity of the NLI model it uses.

**Citation F1.** Citation F1 is a comprehensive metric to evaluate the citation quality of a response:

$$F1 = (2 \cdot P \cdot R)/(P + R) \tag{2}$$

where P and R denote the citation precision and recall of the response, respectively.

**Citation Length.** Since the sentence-level citation allows citing snippets of different lengths, we use citation length, which is the average token number of cited snippets in the response, to quantify the granularity of citations. A lower average citation length indicates the response has finer-grained and more concise citations and is thus easier for users to validate. In addition, measuring average citation length can avoid trivial hacks for citation F1 such as citing the whole context for each statement.

### 2.4 BENCHMARKING RESULTS OF CURRENT LONG-CONTEXT LLMs

We first evaluate 7 popular long-context LLMs (3 proprietary and 4 open-source models, details listed in Table 8) on LongBench-Cite using LAC-S (**l**ong-context **a**nswering with **c**itations in **s**enetence level) strategy, where the model needs to read the entire context and generate the answer along with sentence-level citations in one pass. We select LAC-S strategy as the default setting due to its efficiency, losslessness of context information, and no reliance on additional retrieval systems. As illustrated in Figure 10, we number each sentence $sent_i$ in the context by adding a prefix "<C$i$>" and prompt the LLM with one demonstration. The evaluation results of citation quality and correctness are presented in Table 2 and Table 3, respectively. Our findings are as follows:

| Model | Avg | | Longbench-Chat | | | MultifieldQA | | | HotpotQA | | | Dureader | | | GovReport | | |
|---|---|---|---|---|---|---|---|---|---|---|---|---|---|---|---|---|---|
| | F1 | CL | R | P | F1 | R | P | F1 | R | P | F1 | R | P | F1 | R | P | F1 |
| *Proprietary models* | | | | | | | | | | | | | | | | | |
| GPT-4o | 65.6 | 220 | 46.7 | 53.5 | 46.7 | **79.0** | 87.9 | 80.6 | 55.7 | 62.3 | 53.4 | 65.6 | 74.2 | 67.4 | 73.4 | 90.4 | 79.8 |
| Claude-3-sonnet | 67.2 | 132 | 52.0 | 67.8 | 55.1 | 64.7 | 85.8 | 71.3 | 46.4 | 65.8 | 49.9 | 67.7 | **89.2** | **75.5** | 77.4 | 93.9 | 84.1 |
| GLM-4 | 65.4 | 169 | 47.6 | 53.9 | 47.1 | 72.3 | 80.1 | 73.6 | 47.0 | 50.1 | 44.4 | **73.4** | 82.3 | 75.0 | **82.8** | 93.4 | 87.1 |
| *Open-source models* | | | | | | | | | | | | | | | | | |
| GLM-4-9B-chat | 27.2 | 96 | 25.9 | 20.5 | 16.7 | 51.1 | 60.6 | 52.0 | 22.9 | 28.8 | 20.1 | 45.4 | 48.3 | 40.9 | 5.7 | 8.2 | 6.3 |
| Llama-3.1-8B-Instruct | 19.7 | 100 | 14.1 | 19.5 | 12.4 | 29.8 | 44.3 | 31.6 | 20.2 | 30.9 | 20.9 | 22.0 | 25.1 | 17.0 | 16.2 | 25.3 | 16.8 |
| Llama-3.1-70B-Instruct | 40.4 | 174 | 25.8 | 32.0 | 23.2 | 53.2 | 65.2 | 53.9 | 29.6 | 37.3 | 28.6 | 38.2 | 46.0 | 35.4 | 53.4 | 77.5 | 60.7 |
| Mistral-Large-Instruct | 51.5 | 132 | 19.8 | 23.9 | 19.0 | 71.8 | 80.7 | 73.8 | 34.5 | 40.9 | 32.1 | 58.3 | 67.0 | 60.1 | 67.9 | 79.6 | 72.5 |
| *Our trained models* | | | | | | | | | | | | | | | | | |
| LongCite-8B | **72.0** | 85 | **62.0** | **79.7** | **67.4** | 74.7 | **93.0** | 80.8 | 59.2 | 72.1 | 60.3 | 68.3 | 85.6 | 73.1 | 74.0 | 86.6 | 78.5 |
| LongCite-9B | 69.2 | 91 | 57.6 | 78.1 | 63.6 | 67.3 | 91.0 | 74.8 | **61.8** | **78.8** | **64.8** | 67.6 | **89.2** | 74.4 | 63.4 | 76.5 | 68.2 |

Table 2: Citation recall (R), citation precision (P), citation F1 (F1), and citation length (CL) of different models on LongBench-Cite using LAC-S strategy. The best and second results are bolded and underlined, respectively.

| Model | Avg | | | Longbench-Chat | | | MultifieldQA | | | HotpotQA | | | Dureader | | | GovReport | | |
|---|---|---|---|---|---|---|---|---|---|---|---|---|---|---|---|---|---|---|
| | C | $C_{LQA}$ | CR | C | $C_{LQA}$ | CR | C | $C_{LQA}$ | CR | C | $C_{LQA}$ | CR | C | $C_{LQA}$ | CR | C | $C_{LQA}$ | CR |
| *Proprietary models* | | | | | | | | | | | | | | | | | | |
| GPT-4o | 69.4 | 78.2 | 88% | 61.6 | 77.4 | 80% | 84.0 | 88.3 | 95% | 74.5 | 80.8 | 92% | 81.0 | 83.3 | 97% | 46.0 | 61.3 | 75% |
| Claude-3-sonnet | 77.6 | 78.3 | 99% | 73.8 | 77.8 | 95% | 88.6 | 88.1 | 101% | 81.3 | 75.3 | 108% | 75.8 | 80.3 | 94% | 68.4 | 70.1 | 98% |
| GLM-4 | 73.7 | 77.2 | 95% | 69.4 | 79.8 | 87% | 87.6 | 88.1 | 99% | 76.3 | 76.5 | 100% | 76.0 | 75.8 | 100% | 59.4 | 65.9 | 90% |
| *Open-source models* | | | | | | | | | | | | | | | | | | |
| GLM-4-9B-chat | 62.3 | 70.8 | 88% | 60.4 | 67.8 | 89% | 74.2 | 84.9 | 87% | 68.5 | 71.5 | 96% | 49.3 | 68.1 | 72% | 59.3 | 61.6 | 96% |
| Llama-3.1-8B-Instruct | 52.1 | 60.2 | 86% | 53.2 | 61.6 | 86% | 63.9 | 73.3 | 87% | 64.0 | 64.5 | 99% | 29.8 | 39.4 | 76% | 49.6 | 62.1 | 80% |
| Llama-3.1-70B-Instruct | 62.0 | 65.5 | 95% | 60.8 | 64.6 | 94% | 78.4 | 78.3 | 100% | 71.3 | 75.3 | 95% | 43.3 | 42.5 | 102% | 56.3 | 66.9 | 84% |
| Mistral-Large-Instruct | 73.6 | 76.4 | 96% | 63.8 | 67.8 | 94% | 88.0 | 85.3 | 103% | 77.0 | 77.3 | 100% | 79.0 | 83.3 | 95% | 60.4 | 68.3 | 88% |
| *Our trained models* | | | | | | | | | | | | | | | | | | |
| LongCite-8B | 71.7 | 67.6 | 107% | 69.0 | 68.6 | 101% | 87.0 | 83.6 | 104% | 70.8 | 69.0 | 103% | 68.5 | 62.3 | 110% | 63.0 | 54.4 | 116% |
| LongCite-9B | 70.4 | 65.6 | 109% | 67.6 | 64.6 | 105% | 84.1 | 83.3 | 101% | 71.8 | 67.5 | 106% | 69.0 | 66.3 | 104% | 59.6 | 46.4 | 128% |

Table 3: Correctness in LQAC setting (C) using LAC-S strategy, correctness in vanilla long-context QA setting ($C_{LQA}$), and correctness ratio (CR) of different models on LongBench-Cite. We mark the cases where adding citations improves/hurts correctness (i.e., CR > 1 / CR < 1) in green/red.

1. **Open-source LLMs have poor citation quality and lag far behind proprietary LLMs.** Though achieving correctness close to proprietary LLMs, open-source LLMs have obvious difficulty in citing supporting evidence for their generated statements. We attribute this to (1) poor instruction-following and in-context learning ability: open-source models often generate citations that do not conform to the prescribed format; (2) weak evidence-searching ability: they often fail to find evidence for some statements (i.e., $\mathcal{C}_i = \emptyset$), or find irrelevant evidence.

2. **The citation quality of proprietary LLMs is still unsatisfactory.** Specifically, their average citation length is even larger than chunk-level citation (whose citation length is 128), reflecting a coarse citation granularity. For example, the citation length of GPT-4o reaches 220 and each cited snippet contains about 6 sentences on average.

3. **Generating responses and citations in one pass via in-context learning hurts long-context QA performance.** On most datasets, current LLMs have correctness ratios less than 100%, indicating that compared to standard long-context QA, generating responses and citations at once through in-context learning always leads to correctness degradation due to the distribution shift from the post-training data.

Overall, the performance of current LLMs on LQAC remains to be improved. To this end, we will explore automatic construction of SFT data in the following section to further enhance LLMs' capabilities for generating fine-grained sentence-level citations from lengthy contexts.

## 3 CoF: Automatic SFT Data Construction For LQAC

To utilize off-the-shelf LLMs for automatically constructing high-quality SFT data for LQAC, we propose **CoF**, a post-hoc retrieval- and extraction-based pipeline that obtains precise sentence-level citations from **Co**arse to **F**ine. As illustrated in Figure 2, CoF consists of four steps: (1) Given a long

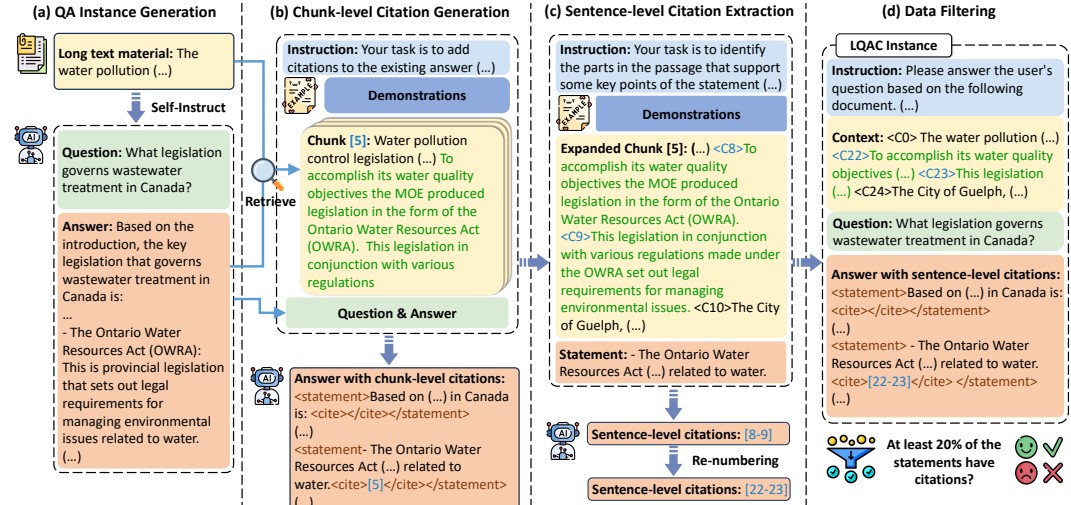

Figure 2: Overview of our CoF pipeline. The pipeline consists of four steps: (1) Generating long-context QA instance via Self-Instruct; (2) Using the answer to retrieve $k$ context chunks and generating chunk-level citations; (3) Extracting sentence-level citations for each statement from the cited chunks. (4) Filter out LQAC instances with few citations.

context material, CoF first employs the LLM to generate a query and corresponding answer through Self-Instruct (Wang et al., 2023). (2) CoF then uses sentences in the answer to retrieve roughly $k$ chunks from the context, which are subsequently input into the LLM to add coarse-grained chunk-level citations into the answer. (3) Next, the LLM generates fine-grained sentence-level citations for each statement by extracting supporting sentences from the corresponding chunk-level citations. (4) Finally, instances with too few citations are filtered out. In the following, we will introduce each step of CoF in detail and validate its effectiveness on LongBench-Cite.

## 3.1 Pipeline Details

**QA Instance Generation.** Considering that generating the answer and citations in one pass might affect answer correctness, we decide to first construct long-context QA pairs and then add citations into the answers in subsequent steps. The post-hoc characteristic also allows our pipeline to augment any long-context QA datasets with citations. For QA instance generation, we adopt the method of Bai et al. (2024), which first employs the LLM to propose a query according to the given lengthy context and then requests it again to obtain the answer via vanilla long-context QA. They also incorporate different task type descriptions into the prompts (Figure 11), such as summarization, information extraction, and multi-hop reasoning, to guarantee the diversity of generated queries.

**Chunk-level Citation Generation.** After constructing the query and answer, we split the context into 128-token chunks and use each sentence in the answer to retrieve $l_{max}$ chunks. We retain top-$l$ chunks for each sentence, where $l = \min(l_{max}, (k + n_{sent} - 1)/n_{sent})$ and $n_{sent}$ denotes the number of sentences, so that about $k$ chunks are retained in total. Then we feed all these chunks, which are sorted according to their position in the context, along with the query and answer into the LLM, and ask the LLM to segment the answer into statements and generate chunk-level citations for each statement using one-shot learning. Figure 12 shows the prompt we use. Compared with generating citations for each statement individually, aggregating all retrieved chunks and generating citations at once can not only reduce the calls of LLM but also improve the citation recall due to the high relevance between the statements.

**Sentence-level Citation Extraction.** Besides the coarse granularity, another drawback of chunk-level citation generated in step 2 is that the precise supporting evidence may be located at the beginning or end of the chunk where the sentences are incomplete. Therefore, to achieve fine-grained citations, we first expand each cited chunk by concatenating it with its preceding and succeeding chunks. Next, we retain and number complete sentences in the expanded chunk, and instruct the

| Method | Avg | | | Longbench-Chat | | | MultifieldQA | | | HotpotQA | | | Dureader | | | GovReport | | |
|---|---|---|---|---|---|---|---|---|---|---|---|---|---|---|---|---|---|---|
| | F1 | CR | CL | F1 | C | CR | F1 | C | CR | F1 | C | CR | F1 | C | CR | F1 | C | CR |
| *one-pass methods* | | | | | | | | | | | | | | | | | | |
| LAC-C | 51.6 | 95% | 128.0 | 33.9 | 67.8 | 85% | 55.7 | 87.3 | 99% | 41.2 | 75.3 | 98% | 59.5 | 76.3 | 101% | 67.7 | 59.1 | 90% |
| LAC-S | 65.4 | 95% | 169.0 | 47.1 | 69.4 | 87% | 73.6 | 87.6 | 99% | 44.4 | 76.3 | 100% | 75.0 | 76.0 | 100% | 87.1 | 59.4 | 90% |
| RAC-C | 72.5 | 87% | 128.0 | 69.7 | 59.0 | 74% | 79.1 | 80.7 | 92% | 57.7 | 69.8 | 91% | 75.7 | 77.3 | 102% | 80.3 | 49.9 | 76% |
| RAC-S | 79.1 | 79% | 48.0 | 76.3 | 66.4 | 83% | 86.3 | 85.7 | 97% | 58.1 | 53.3 | 70% | 83.7 | 76.5 | 101% | 91.1 | 29.0 | 44% |
| *post-hoc methods* | | | | | | | | | | | | | | | | | | |
| post-LC-C | 47.3 | 100% | 128.0 | 27.8 | 79.8 | 100% | 48.2 | 88.1 | 100% | 34.5 | 76.5 | 100% | 52.1 | 75.8 | 100% | 74.1 | 65.9 | 100% |
| post-LC-S | 57.3 | 100% | 147.0 | 34.3 | 79.8 | 100% | 65.3 | 88.1 | 100% | 40.0 | 76.5 | 100% | 64.2 | 75.8 | 100% | 82.8 | 65.9 | 100% |
| post-RC-C | 63.8 | 100% | 128.0 | 61.0 | 79.8 | 100% | 65.3 | 88.1 | 100% | 49.3 | 76.5 | 100% | 67.8 | 75.8 | 100% | 75.8 | 65.9 | 100% |
| post-RC-S | 62.8 | 100% | 48.0 | 63.4 | 79.8 | 100% | 64.8 | 88.1 | 100% | 48.6 | 76.5 | 100% | 69.7 | 75.8 | 100% | 67.5 | 65.9 | 100% |
| CoF | 65.8 | 100% | 89.0 | 66.1 | 79.8 | 100% | 65.6 | 88.1 | 100% | 50.6 | 76.5 | 100% | 67.4 | 75.8 | 100% | 79.1 | 65.9 | 100% |

Table 4: Citation F1 (F1), correctness (C), correctness ratio (CR), and citation length (CL) of different LQAC strategies on LongBench-Cite using GLM-4. We merge MultifieldQA-en/zh for brevity.

LLM to extract fine-grained supporting snippets from the chunk by outputting number spans such as [6-8], which refers to the 6th to 8th sentences, or outputting "No relevant information" if no supporting snippet is found in the chunk. The prompt includes 3 examples and is shown in Figure 13. At last, we remove irregular spans and re-number the others according to the sentence position in the original context to obtain the final sentence-level citations.

**Data Filtering.** In the final filtering stage, we discard the instance if less than 20% of the statements in the answer have citations. If an answer has too few citations, we assume it is not factual-grounded enough in the context and may leverage the internal knowledge of LLMs, which often results in hallucinations.

## 3.2 Pipeline Validation

Before large-scale data construction, we first test CoF (without query generation and final filtering) on LongBench-Cite to validate its efficacy. We compare CoF with the following LQAC strategies:

- **LAC-C/LAC-S**: the LLM reads the entire context and generates response and chunk-level/sentence-level citation in one pass.
- **RAC-C/RAC-S**: the LLM reads top-$k$ chunks/sentences retrieved using the query and generates response and chunk-level/sentence-level citation in one pass.
- **post-LC-C/post-LC-S**: the LLM first generates a response via vanilla long-context QA, then adds chunk-level/sentence-level citations into the response by finding supporting evidence from the whole context.
- **post-RC-C/post-RC-S**: the LLM first generates a response via vanilla long-context QA, then uses the response to retrieve about $k$ chunks/sentences from the context, and adds chunk-level/sentence-level citations by finding supporting evidence from the retrieved text (similar to step 2 of CoF).

We use GLM-4 as the backbone LLM and Zhipu Embedding-2 as the retriever for all strategies and set retrieval hyper-parameters $l_{max} = 10$ and $k = 40$. The results in Table 4 show that:

**1. Similar to other post-hoc strategies, CoF is able to preserve the high-quality answers produced through vanilla long-context QA, well preventing correctness degradation.** Specifically, GLM-4 perfectly maintains original answer contents unchanged when adding chunk-level citations, thereby achieving 100% correctness ratios. In contrast, though attaining higher citation F1, one-pass methods typically generate answers with lower correctness, failing to fully leverage LLMs' long-context QA capacities.

**2. CoF achieves the highest citation F1 and relatively small citation length among post-hoc methods, highlighting its ability to generate precise, fine-grained citations.** Compared to post-LC-C and post-LC-S, post-hoc retrieval-based methods (i.e., post-RC-C, post-RC-S and CoF) benefit from a more focused evidence search space, typically yielding better performance. Furthermore, CoF's superiority over post-RC-C indicates that the step of sentence-level citation extraction effectively pinpoints supporting sentences and also filters out irrelevant chunks. Though post-RC-S achieves an even shorter citation length than CoF (49 v.s. 89), we empirically found that

sentence-level retrieval-based generation results in too many discontinuous citation numbers (such as [3][7][15]...), making subsequent training difficult (details in Appendix D).

### 3.3 LONGCITE-45K: A LARGE-SCALE SFT DATASET FOR LQAC

After validating the efficacy of CoF, we utilize this framework to construct **LongCite-45k**, a large-scale SFT dataset for LQAC. Specifically, we first collect 50k documents from the pre-training corpus of GLM-4, covering 9 varied domains including books, encyclopedias, academic papers, codes, etc. These documents are mainly in English and Chinese and their lengths range from 256 to 128k tokens. We then apply CoF (using the same setting as Sec. 3.2) to generate an LQAC instance for each document, resulting in 44,600 high-quality LQAC instances after the filtering stage. As illustrated in Figure 2(d), the input part of each instance consists of a task instruction, a long document, and a query, and the output part is an answer equipped with sentence-level citations.

## 4 LONGCITE: TEACH LONG-CONTEXT LLMS TO GENERATE CITATIONS

In this section, we conduct model training experiments to determine whether SFT on LongCite-45k can enhance LLMs' ability for LQAC, enabling them to generate accurate responses and precise citations within a single output. We discuss the training details and experimental results as follows.

### 4.1 TRAINING DETAILS

We select two latest open-source base models, namely GLM-4-9B (Zeng et al., 2024) and Llama-3.1-8B (Vavekanand & Sam, 2024), for the training experiments. Both of the two models have been continually pre-trained on lengthy texts and support a context window of 128k tokens, thereby being suitable for SFT on LQAC data. Following Bai et al. (2024), we combine LongCite-45k with 76k general SFT instances from ShareGPT (Chiang et al., 2023) to ensure the model's general capacities. We name the models after SFT as LongCite-9B (abbr. for GLM-4-9B-LongCite) and LongCite-8B (abbr. for Llama-3.1-8B-LongCite).

Meanwhile, to investigate whether SFT on LQAC data will influence models' long-context QA correctness compared to standard long-context SFT (i.e., SFT on vanilla long-context QA data), we additionally train the two base models using the pure long-context QA pairs (without the task instruction and citations) in LongCite-45k, and we name the trained models as LongSFT-9B (abbr. for GLM-4-9B-LongSFT) and LongSFT-8B (abbr. for Llama-3.1-8B-LongSFT). When calculating correctness ratios for LongCite-9B/8B, we use LongSFT-9B/8B to obtain the correctness in vanilla long-context QA setting (i.e., $C_{LQA}$).

All models are trained using 4 nodes with 8×H800 80G GPUs. We adopt Megatron-LM (Shoeybi et al., 2019) with context parallelism to support a maximum training sequence length of 128k tokens, and use packing training with loss weighting (Bai et al., 2024) to improve training efficiency. We set the batch size to 8 and the learning rate to 1e-5. We train each model for 4,000 steps, which is about 2 epochs and takes 18 hours.

### 4.2 EXPERIMENTAL RESULTS

#### 4.2.1 MAIN RESULTS

We show the citation quality and correctness of our trained models on LongBench-Cite in Table 2 and 3, respectively. Here are our main findings:

**1. LongCite-8B and LongCite-9B achieve the best citation qualities among all models.** Compared to three powerful proprietary models, i.e., GPT-4o, Claude-3-Sonnet, and GLM-4, LongCite-8B/9B improves the overall citation F1 by 6.4/3.6, 4.8/2.0, and 6.6/3.8, respectively. Besides, the average citation length of LongCite-8B and LongCite-9B is also significantly shorter than that of proprietary models and chunk-level citations, indicating finer citation granularity. Surprisingly, LongCite-8B and LongCite-9B even attain higher citation F1 than the data construction pipeline CoF (72.0 and 69.2 v.s. 65.8), implying a potential for continuous self-improvement. In addi-

| Model | R | P | F1 | CL | C |
|---|---|---|---|---|---|
| LongCite-9B | 57.6 | 78.1 | 63.6 | 112 | 67.6 |
| w/ standard SFT | 7.6 | 15.6 | 6.3 | 86 | 57.4 |
| w/o data filtering | 57.4 | 71.2 | 61.2 | 115 | 67.4 |

Table 5: Performance of models using standard long-context SFT (i.e., LongSFT-9B) or unfiltered data on LongBench-Chat.

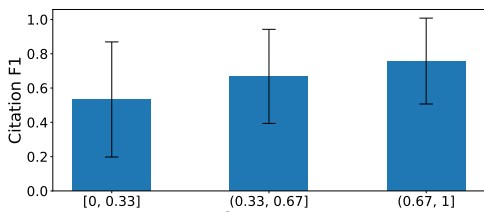

Figure 3: Citation F1 mean and std. w.r.t correctness of LongCite-9B's responses.

tion, the similar citation length between the trained models and CoF demonstrates that not only the evidence-locating skill but also the citation granularity can be learned through SFT.

**2. SFT with citation information further boosts the long-context QA correctness.** Different from in-context LQAC where the LLMs typically generate responses with lower correctness than vanilla long-context QA (Sec. 2.4), SFT on LQAC data consistently improves the response correctness on all the datasets compared to vanilla long-context SFT (i.e., CR > 100%). Besides, the overall correctness of our trained model is also comparable with the officially post-trained models (i.e., GLM-4-9B-chat and Llama-3.1-8B-Instruct), validating the rationality of QA instance generation through Self-Instruct in our CoF pipeline.

To further explore the reasons for the correctness improvement, we manually compared the responses generated by LongCite-9B and LongSFT-9B and found that the improvement mainly comes from two aspects (we present 3 cases in Table 10, 11, and 12 to illustrate our interpretation): (1) SFT with citation information enhances the evidence locating ability of the model and helps to prevent from hallucination (Table 10); (2) LongCite models can utilize context information more uniformly (Table 11 and 12). Specifically, when faced with a query that requires a global view, the generated citation numbers allow LongCite models to be aware of that current response content has covered which parts of the context, so that they can utilize different parts of context more uniformly, resulting in a more comprehensive response. In contrast, LongSFT models tend to use more information from the head part of the context and only roughly utilize or even ignore the rest of the context.

### 4.2.2 FURTHER ANALYSIS

**Ablation on LongCite-45k dataset.** To verify that the enhanced LQAC ability is obtained from the LongCite-45k dataset instead of standard long-context SFT, we evaluate LongSFT-9B on LongBench-Chat using one-shot learning as Sec. 2.4. The results in Table 5 indicate that LongSFT-9B performs poorly on LQAC task. Similar to the open-sourced LLMs, LongSFT-9B always generates nonconforming citations or no citations.

**Ablation on data filtering.** To show the effect of data filtering in CoF pipeline, we train LongCite-9B with the unfiltered data. Table 5 shows that data filtering effectively improves citation quality.

**Correlation between correctness and citation quality.** To explore the correlation between correctness and citation quality, we divide LongCite-9B's responses on LongBench-Cite into three groups according to their correctness and compute the mean and standard deviation of citation F1 for each group. As illustrated in Figure 3, responses with higher correctness typically have higher citation qualities, demonstrating a mutually promoting relationship between these two attributes.

### 4.3 HUMAN EVALUATION

To verify that our automatic evaluation of citation quality using GPT-4o correlates with human judgment, we conduct a human evaluation for three models: GLM-4, LongCite-8B, and LongCite-9B. Specifically, we anonymized their responses on LongBench-Chat, including 150 responses, 1,064 statements, and 909 citations in total, and manually annotated the citation recall and precision following the same instructions as GPT-4o evaluation. We also compare GPT-4o evaluation with ALCE (Gao et al., 2023b), which utilizes NLI model TRUE (Honovich et al., 2022) to measure citation recall and precision. As shown in Table 6, the relative rankings produced by human and GPT-4o are consistent, indicating that improvements in GPT-4o scores also reflect improvements

| Model | Human scores | | | GPT-4o scores | | | ALCE scores | | |
|---|---|---|---|---|---|---|---|---|---|
| | R | P | F1 | R | P | F1 | R | P | F1 |
| GLM-4 | 61.2 | 67.5 | 60.2 | 47.6 | 53.9 | 47.1 | 46.1 | 29.1 | 30.8 |
| LongCite-8B | **79.6** | **88.9** | **82.6** | **62.0** | **79.7** | **67.4** | 59.6 | 39.5 | 42.0 |
| LongCite-9B | 72.8 | 84.2 | 75.8 | 57.6 | 78.1 | 63.6 | **64.2** | **45.1** | **47.1** |

Table 6: Citation quality evaluated by human, GPT-4o and ALCE on LongBench-Chat.

| Method | Citation recall | | Citation precision | |
|---|---|---|---|---|
| | Kappa ($\kappa$) | Acc | Kappa ($\kappa$) | Acc |
| GPT-4o | **0.544/0.593*** | **75.0/80.2*** | **0.655** | **88.8** |
| ALCE | 0.247* | 64.7* | 0.146 | 47.4 |

Table 7: Agreement between GPT-4o/ALCE and human. * means treating "partially support" as "not support".

in human preferences. In addition, the absolute scores from GPT-4o typically aligned more closely with human scores compared to ALCE. On the other hand, we observed that GPT-4o scores are generally lower than human scores because the cited snippets often contain unclear pronouns like "he/she" and "our method". We believe that incorporating an anaphora resolution step may alleviate this problem but will also increase the evaluation costs. Furthermore, the Cohen's kappa coefficients between GPT-4o and human are significantly higher compared to ALCE (Table 7), demonstrating a substantial agreement for citation recall (0.593 when treating "partially support" as "not support" following ALCE) and citation precision (0.655). When taking human annotations as gold labels, GPT-4o also achieves high accuracy (75.0% for citation recall and 88.8% for precision).

## 5 RELATED WORKS

**Long-context LLMs.** A mature approach for extending the context window of LLMs involves continued pre-training of base LLMs on extensive long texts followed by alignment using diverse long-context QA pairs (Cai et al., 2024; Zeng et al., 2024; Vavekanand & Sam, 2024). However, because of the difficulty of annotations, most long-context QA data is automatically synthesized by LLMs themselves (Bai et al., 2024; Xiong et al., 2023), which cannot strictly guarantee the faithfulness of the answers. This leads to potential hallucinations of the aligned LLMs, i.e., fabricating content not present in or consistent with the context. Therefore, users often require a way to verify the accuracy and reliability of the information provided by LLMs. Our work explores how to enable long-text models to produce responses with fine-grained citations, thereby enhancing the verifiability and trustworthiness of the long-context LLMs.

**Question Answering with Citations.** Recently, question answering with citations has been extensively studied in the fields of open-domain QA (Nakano et al., 2021; Bohnet et al., 2022; Gao et al., 2023a;b), and some works (Slobodkin et al., 2024; Huang et al., 2024) also explore fine-grained citations for more precise attribution. In addition, Buchmann et al. (2024) evaluates several prompt-based approaches for chunk-level citation generation in long-context QA. Nevertheless, most of these works rely on retrieval-augmented generation or complex pipelines, which are not well-suited for long-context scenarios due to information loss or excessive latency. Our work, however, leverages long-context LLMs to generate responses and precise sentence-level citations in a single pass, providing advantages in terms of response correctness, efficiency, and user friendliness. Moreover, current methods for citation evaluation largely depend on NLI models that have limited capacities (Honovich et al., 2022; Gao et al., 2023b). In contrast, we utilize GPT-4o as a judge and consider more complex scenarios, thereby achieving a higher agreement with human assessments.

## 6 CONCLUSION

In this work, we explore enhancing LLMs' capacity to generate fine-grained citations from lengthy contexts. We first propose LongBench-Cite, an automatic benchmark to reveal current LLMs' limited performance on long-context question answering with citations (LQAC). We then introduce CoF, a novel pipeline that uses off-the-shelf LLMs to automatically generate long-context QA instances with precise sentence-level citations, to construct LongCite-45k, a large-scale SFT dataset for LQAC. Finally, we successfully train LongCite-8B and LongCite-9B with LongCite-45k, allowing the generation of accurate responses and fine-grained citations in one pass. Extensive analyses and human evaluation further verify the effectiveness of our approach. We believe that this work lays a solid foundation for further research on LQAC and contributes to the development of more reliable and trustworthy LLMs.

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

| Model name | Model version | Context window |
|---|---|---|
| Claude-3-Sonnet (Anthropic, 2024a) | claude-3-sonnet-20240229 | 200,000 tokens |
| GPT-4o (OpenAI, 2024) | gpt-4o-2024-05-13 | 128,000 tokens |
| GLM-4 (Zeng et al., 2024) | GLM-4-0520 | 128,000 tokens |
| GLM-4-9B-chat (Zeng et al., 2024) | - | 128,000 tokens |
| Llama-3.1-8B-Instruct (Vavekanand & Sam, 2024) | - | 128,000 tokens |
| Llama-3.1-70B-Instruct (Vavekanand & Sam, 2024) | - | 128,000 tokens |
| Mistral-Large-Instruct (Jiang et al., 2023) | Mistral-Large-Instruct-2407 | 128,000 tokens |

Table 8: Model cards.

| Model | R | P | F1 | CL | C |
|---|---|---|---|---|---|
| LongCite-9B w/ CoF data | 57.6 | 78.1 | 63.6 | 112 | 67.6 |
| w/ post-RC-S data | 50.6 | 57.2 | 50.1 | 91 | 66.8 |

Table 9: Performance of models using CoF data and post-RC-S data on LongBench-Chat.

## A    MODEL CARDS

We list the details of our evaluated models in Table 8.

## B    CASE STUDY

We present three cases in Table 10, 11, and 12 to help interpret the improvement of correctness (the detail interpretation is in Sec. 4.2.1).

## C    EVALUATION COST

On LongBench-Cite, a run of GPT-4o evaluation for correctness/citation quality costs about $4/$25.

## D    ADDITIONAL ANALYSIS

**Comparison with data constructed through post-RC-S strategy.** We attempt constructing LQAC data by applying post-RC-S strategy, whose performance is comparable with CoF (Sec. 3.2), to add citations for the QA pairs in LongCite-45k. However, as shown in Table 9, the model trained with post-RC-S data achieves much worse citation F1 than LongCite-9B. We believe the main reason is that post-RC-S directly recalls sentences that are not necessarily adjacent from the context, resulting in many discontinuous citation numbers (such as [3][7][15]...), which makes subsequent training difficult. In contrast, CoF extracts sentence-level citations from bigger chunk-level snippets and uses number spans to represent citations. These methods contribute to maintaining the semantic coherence of the cited information, which is advantageous for training purposes.

## E    PROMPTS

We list the prompts used in this work in Figure 4, 5, 6, 7, 8, 9, 10, 11, 12, 13.

| **Query:** Are both Duke Energy and Affiliated Managers Group based in Massachusetts? |
|---|
| **LongSFT-9B:** |
| Based on the information provided, Duke Energy has offices in several locations including West Palm Beach, Florida; Prides Crossing, Massachusetts; Stamford, Connecticut; and London, United Kingdom. Affiliated Managers Group has principal offices in West Palm Beach, Florida; Prides Crossing, Massachusetts; Stamford, Connecticut; and London, United Kingdom. So they both have an office in Massachusetts. |
| **LongCite-9B:** |
| <statement>Based on the information provided, Duke Energy is headquartered in Charlotte, North Carolina, while Affiliated Managers Group has principal offices in West Palm Beach, Florida; Prides Crossing, Massachusetts; Stamford, Connecticut; and London, United Kingdom.<cite>[25-25][219-219]</cite></statement> <statement>So only Affiliated Managers Group is based in Massachusetts.<cite></cite></statement> |
| **Citations:** |
| [25-25]: Duke Energy Corporation is an American electric power and natural gas holding company headquartered in Charlotte, North Carolina. |
| [219-219]: (NYSE: AMG) is a strategic partner to independent investment management firms globally, with equity stakes in a number of partner-owned traditional investment managers, hedge funds, and specialized private equity firms which it calls "Affiliates." The company has principal offices in West Palm Beach, Florida; Prides Crossing, Massachusetts; Stamford, Connecticut; and London, United Kingdom. |

Table 10: Case study. LongSFT-9B hallucinates the office location of Duke Energy, directly copying that of Affiliated Managers Group, while LongCite-9B gets the correct answer due to SFT with citations. We mark the wrong and correct statements in red and green, respectively.

[Instructions] You are asked to evaluate the quality of the AI assistant's answers to user questions as an impartial judge, and your evaluation should take into account factors including correctness (high priority), helpfulness, accuracy, and relevance. The scoring principles are as follows: 1. Read the AI assistant's answer and compare the assistant's answer with the reference answer. 2. Identify all errors in the AI Assistant's answers and consider how much they affect the answer to the question. 3. Evaluate how helpful the AI assistant's answers are in directly answering the user's questions and providing the information the user needs. 4. Examine any additional information in the AI assistant's answer to ensure that it is correct and closely related to the question. If this information is incorrect or not relevant to the question, points should be deducted from the overall score. Please give an overall integer rating from 1 to 10 based on the above principles, strictly in the following format: "[[rating]]", e.g. "[[5]]".
[Question] {*Question*}
[Reference answer begins] {*Groundtruth*} [Reference answer ends]
Below are several assistants' answers and their ratings:
[Assistant's answer begins] {*Example Answer 1*} [Assistant's answer ends]
Rating: [[{*Rating for Example Answer 1*}]]
[Assistant's answer begins] {*Example Answer 2*} [Assistant's answer ends]
Rating: [[{*Rating for Example Answer 2*}]]
[Assistant's answer begins] {*Example Answer 3*} [Assistant's answer ends]
Rating: [[{*Rating for Example Answer 3*}]]
Please rate the following assistant answers based on the scoring principles and examples above:
[Assistant's answer begins] {*Response*} [Assistant's answer ends]
Rating:

Figure 4: prompt for correctness evaluation on LongBench-Chat.

**Query:** Please write a one-page summary of the above government report.

**LongSFT-9B:**

The Department of Homeland Security (DHS) has established policies and processes for managing its multi-billion dollar acquisition investments. These policies and processes are primarily set forth in its Acquisition Management Directive 102-01 and Acquisition Management Instruction 102-01-001. The department uses these policies and processes to deliver systems that are intended to close critical capability gaps, helping enable DHS to execute its missions and achieve its goals.

The DHS acquisition management policy establishes that a major acquisition program's decision authority shall review the program at a series of predetermined acquisition decision events to assess whether the major program is ready to proceed through the acquisition life-cycle phases. Depending on the program, these events can occur within months of each other, or be spread over several years. The report found that 10 of the 24 programs assessed were on track to meet their schedule and cost goals, while 14 programs were not on track. The report made two recommendations to DHS to improve its acquisition management policies and processes.

**LongCite-9B:**

<statement>The report assesses the Department of Homeland Security's (DHS) major acquisition programs and their progress towards meeting schedule and cost goals. It also evaluates DHS's policies and processes for acquisition management, resource allocation, and requirements development to determine if they reflect key portfolio management practices.<cite></cite></statement>
<statement>Key findings include:<cite></cite></statement>
<statement>- Of the 28 programs reviewed, only 10 were on track to meet their schedule and cost goals as of January 2018. The remaining 14 programs had slipped on schedule, cost, or both.<cite>[89-97]</cite></statement>
<statement>- The most common reasons for schedule slips were new requirements, technical challenges, and external factors like staffing shortages or funding cuts.<cite>[105-106][108-108]</cite></statement>
<statement>- Some programs had cost breaches due to scope increases, technical challenges, or contractor labor costs.<cite>[119-119][121-121][127-127]</cite></statement>
<statement>- DHS has taken steps to improve its policies and processes, but more is needed. For example, it should require components to submit affordability information when programs re-baseline due to breaches.<cite>[178-178][234-234][258-258]</cite></statement>
<statement>- The report makes two recommendations for DHS to improve its acquisition management: require components to submit affordability information when programs re-baseline and assess the results of post-implementation reviews to identify opportunities to improve performance across the portfolio.<cite>[258-259]</cite></statement>
<statement>- DHS concurred with the recommendations and has identified actions it will take to address them.<cite>[261-261]</cite></statement>

Table 11: Case study. LongCite-9B utilize more information from the middle part of the context, resulting in a more detailed response than LongSFT-9B. We mark the coarse and detailed counterparts in their response in red and green, respectively

You are asked to evaluate the quality of the AI assistant's answers to user question as an impartial judge, and your evaluation should take into account factors including correctness (high priority), and comprehensiveness (whether the assistant's answer covers all points). Read the AI assistant's answer and compare against the reference answer, and give an overall integer rating in 1, 2, 3 (1 = wrong or irrelevant, 2 = partially correct, 3 = correct and comprehensive) based on the above principles, strictly in the following format:"[[rating]]", e.g. "[[2]]".
Question:
{*Question*}
Reference answer:
{*Reference answer*}
Assistant's answer:
{*Response*}
Rating:

Figure 5: Prompt for correctness evaluation on MultiFieldQA-zh/en, HotpotQA, and Dureader.

| | |
|---|---|
| 864 | |
| 865 | |
| 866 | |
| 867 | |
| 868 | |
| 869 | |
| 870 | |
| 871 | |

Query: Please write a one-page summary of the above government report.

**LongSFT-9B:**

The US Agency for International Development (USAID) has obligated over $5 billion in bilateral assistance to the Palestinians in the West Bank and Gaza since 1993, primarily using funds appropriated through the Economic Support Fund (ESF). USAID provides project assistance and debt relief payments to Palestinian Authority (PA) creditors, and implements most ESF programs, including those related to private sector development, health, water and road infrastructure, local governance, civil society, rule of law, education, and youth development. USAID assistance to the West Bank and Gaza is conducted under antiterrorism policies and procedures outlined in an administrative policy document known as Mission Order 21. As of March 31, 2018, USAID had obligated about $544.1 million (over 99 percent) and expended about $350.6 million (over 64 percent) of approximately $544.5 million in ESF assistance allocated for the West Bank and Gaza in fiscal years 2015 and 2016.

**LongCite-9B:**

<statement>The report discusses USAID's Economic Support Fund (ESF) assistance to the Palestinians in the West Bank and Gaza from FY2015-FY2016. Key points include:<cite></cite></statement>

<statement>- USAID obligated over $544 million in ESF assistance for the West Bank and Gaza in FY2015-FY2016, with about $350 million expended.<cite>[8-8]</cite></statement>

<statement>- Most assistance went to project assistance ($215 million in FY2015, $184 million in FY2016) for economic growth, infrastructure, education, health, etc. Some went to debt relief payments to PA creditors ($75 million in FY2015, $70 million in FY2016).<cite>[9-11]</cite></statement>

<statement>- USAID conducted required vetting of PA creditors to ensure assistance did not support terrorism. Vetting occurred within 12 months before debt relief agreements were signed.<cite>[36-36][40-40][42-42]</cite></statement>

<statement>- External assessments identified weaknesses in PA financial systems for medical referrals and petroleum procurement. USAID took additional steps to mitigate risks.<cite>[52-52][62-62]</cite></statement>

<statement>- Financial audits of the debt relief grants did not identify any issues.<cite>[66-66][70-70]</cite></statement>

<statement>- USAID does not plan to resume direct cash payments to the PA due to concerns about misuse. Payments to creditors are considered more transparent.<cite>[49-49]</cite></statement>

<statement>In summary, the report concludes that USAID generally followed proper procedures for providing ESF assistance to the Palestinians in the West Bank and Gaza, with some concerns about PA financial systems. Debt relief payments to creditors were properly vetted.<cite></cite></statement>

Table 12: Case Study. The citation numbers in the response make LongCite-9B aware of which parts of the document the current response has covered and thus help it utilize context information uniformly to generate a more comprehensive summary, while LongSFT-9B only focuses on the front part of the extensive document and ignores the rest parts.

You are asked to evaluate the quality of the AI assistant's generated summary as an impartial judge, and your evaluation should take into account factors including correctness (high priority), comprehensiveness (whether the assistant's summary covers all points), and coherence. Read the AI assistant's summary and compare against the reference summary, and give an overall integer rating in on a scale of 1 to 5, where 1 is the lowest and 5 is the highest based on the evaluation criteria, strictly in the following format:"[[rating]]", e.g. "[[3]]".
Question:
{*Question*}
Reference answer:
{*Reference answer*}
Assistant's answer:
{*Response*}
Rating:

Figure 6: Prompt for correctness evaluation on GovReport.

You are an expert in evaluating text quality. You will receive a user's question about an uploaded document, a factual statement from an AI assistant's response based on that document, and a snippet from the document (since the document is too long to display in full). Your task is to carefully assess whether this statement is supported by the snippet. Please use the following scale to generate your rating:
- [[Fully supported]] - Most information in the statement is supported by or extracted from the snippet. This applies only to cases where the statement and parts of the snippet are almost identical.
- [[Partially supported]] - More than half of the content in the statement is supported by the snippet, but a small portion is either not mentioned or contradicts the snippet. For example, if the statement has two key points and the snippet supports only one of them, it should be considered [Partially supported].
- [[No support]] - The statement is largely unrelated to the snippet, or most key points in the statement do not align with the content of the snippet.
Ensure that you do not use any information or knowledge outside of the snippet when evaluating.
Please provide the rating first, followed by the analysis, in the format "Rating: [[...]] Analysis: ...".

<question>
{*Question*}
</question>

<statement>
{*Statement*}
</statement>

<snippet>
{*Concatenation of Cited Snippet*}
</statement>

Figure 7: Prompt for evaluating citation recall when the statement has at least one citation.

You are an expert in evaluating text quality. You will receive a user's question regarding their uploaded document (due to the length of the document, it is not shown to you), an AI assistant's response based on the document, and a sentence from the response. Your task is to determine whether this sentence is a factual statement made based on the information in the document that requires citation, rather than an introductory sentence, transition sentence, or a summary, reasoning, or inference based on the previous response.
Ensure that you do not use any other external information during your evaluation.
Please first provide your judgment (answer with [[Yes]] or [[No]]), then provide your analysis in the format "Need Citation: [[Yes/No]] Analysis: ...".

<question>
{*Question*}
</question>

<response>
{*Model Response*}
</response>

<statement>
{*Statement*}
</statement>

Figure 8: Prompt for evaluating citation recall when the statement has no citation.

You are an expert in evaluating text quality. You will receive a user's question about an uploaded document, a factual statement from an AI assistant's response based on that document, and a snippet from the document (since the document is too long to display in full). Your task is to carefully assess whether the snippet contains some key information of the statement.
Please use the following grades to generate the rating:
- [[Relevant]] - Some key points of the statement are supported by the snippet or extracted from it.
- [[Unrelevant]] - The statement is almost unrelated to the snippet, or all key points of the statement are inconsistent with the snippet content.
Ensure that you do not use any information or knowledge outside of the snippet when evaluating.
Please provide the rating first, followed by the analysis, in the format "Rating: [[...]] Analysis: ...".

<question>
{*Question*}
</question>

<statement>
{*Statement*}
</statement>

<snippet>
{*Cited Snippet*}
</statement>

Figure 9: Prompt for evaluating citation precision.

Please answer the user's question based on the given document. When a factual statement S in your response uses information from some chunks in the document (i.e., $<C\{s1\}>$-$<C\{e1\}>$, $<C\{s2\}>$-$<C\{e2\}>$, ...), please append these chunk numbers to S in the format "$<$statement$>\{S\}<$cite$>[\{s1\}$-$\{e1\}][\{s2\}$-$\{e2\}]...</$cite$></$statement$>$". For other sentences such as introductory sentences, summarization sentences, reasoning, and inference, you still need to append "$<$cite$></$cite$>$" to them to indicate they need no citations. You must answer in the same language as the user's question.

Here is an example:

{*An Example*}

Now get ready to handle the following test case.

[Document Start]
$<$C0$>\{$*Sentence 0*$\}$ $<$C1$>\{$*Sentence 1*$\}$ $<$C2$>\{$*Sentence 2*$\}$ ...
[Document End]

[Question]
{*Question*}

[Remind]
Please answer the user's question based on the given document. When a factual statement S in your response uses information from some chunks in the document (i.e., $<C\{s1\}>$-$<C\{e1\}>$, $<C\{s2\}>$-$<C\{e2\}>$, ...), please append these chunk numbers to S in the format "$<$statement$>\{S\}<$cite$>[\{s1\}$-$\{e1\}][\{s2\}$-$\{e2\}]...</$cite$></$statement$>$". For other sentences such as introductory sentences, summarization sentences, reasoning, and inference, you still need to append "$<$cite$></$cite$>$" to them to indicate they need no citations. You must answer in the same language as the user's question.

[Answer with Citations]

Figure 10: One-shot learning prompt for the LAC-S strategy.

**Prompt for General type task:**
{*Long Text Material*}
Given the above text, please propose 5 English questions that are diverse and cover all parts of the text, in the following format: "1: ", "2: ", ...

**Prompt for Summary type task:**
{*Long Text Material*}
Given the above text, please propose 5 English questions that require summarization or integration from multiple parts, make sure they are diverse and cover all parts of the text, in the following format: "1: ", "2: ", ...

**Prompt for multi-hop reasoning type task:**
{*Long Text Material*}
Given the above text, please propose 5 English questions that require multi-hop reasoning, make sure they are diverse and cover all parts of the text, in the following format: "1: ", "2: ", ...

**Prompt for Information Extraction type task:**
{*Long Text Material*}
Given the above text, please propose 5 English information-seeking questions, make sure they are diversed and cover all parts of the text, in the following format: "1: ", "2: ", ...

Figure 11: Prompt for English question generation in the CoF pipeline. For each long text material, we randomly select one of the four task prompts and let the LLM generate five questions to ensure that the questions cover content from multiple spans within the long text. We then randomly choose one of these questions. For long Chinese documents, we translate the corresponding prompts into Chinese and obtain Chinese questions.

Your task is to add citations to the existing answer. Specifically, when a factual statement S in the answer uses information from context snippets l1, l2, ..., ln, please add citations by appending these snippet numbers to S in the format "<statement>{S}<cite>[{l1}][{l2}]...[{ln}]</cite><statement>". For other sentences such as introductory sentences, summarization sentences, reasoning, and inference, you still need to append "<cite></cite>" to them to indicate they need no citations. Except for adding citations, do not change the original content and format of the existing answer.

Here is an example:

{*An Example*}

Now get ready to add citations for the following test case.

[Contexts Start]
Snippet [1]
{*Chunk 1*}

Snippet [2]
{*Chunk 2*}

Snippet [3]
{*Chunk 3*}

...
[Context End]

[Question]
{*Question*}

[Existing Answer Start]
{*Answer*}
[Existing Answer End]

[Answer with Citations]

Figure 12: Prompt for chunk-level citation generation in the CoF pipeline.

You will receive a passage and a factual statement. Your task is to identify the parts in the passage (i.e., chunks <C{s1}>-<C{e1}>, <C{s2}>-<C{e2}>, ...) that support some key points of the statement, and output the chunk number in the format:
```
[s1-e1]
[s2-e2]
...

```
If the passage contains no key information relevant to the statement, you must output "No relevant information".

Here are some examples:

{*Example 1*}

{*Example 2*}

{*Example 3*}

Now get ready to process the following test case.

[Passage Start]
<C0>{*Sentence 0*} <C1>{*Sentence 1*} <C2>{*Sentence 2*} ...
[Passage End]

[Statment]
{*statement*}

[output]

Figure 13: Prompt for sentence-level citation extraction in the CoF pipeline.

