# OpenReview forum: "LongCite: Enabling LLMs to Generate Fine-grained Citations in Long-context QA"
_ICLR.cc/2025/Conference — ICLR 2025 Conference Withdrawn Submission_

### Official Review · Reviewer_UmK2 · 2024-10-28

**Soundness:** 2
**Presentation:** 2
**Contribution:** 2
**Rating:** 3
**Confidence:** 2

**Summary:**

This paper focuses on the Long-Context Question Answering with Citations (LQAC) task. To assess large language models (LLMs) in generating accurate answers with appropriate sentence-level citations, the authors introduce an automated benchmark called **LongBench-Cite**. They further develop a supervised fine-tuning dataset **LongCite-45k** using a **Coarse to Fine (CoF) pipeline**. Leveraging this SFT dataset, they train **two LLMs** that achieve state-of-the-art citation quality on their curated benchmark.

**Strengths:**

- The paper is well-structured, effectively integrating the benchmark, training data, and fine-tuned models for the LQAC task.

- The evaluation results on LongBenchCite demonstrate that the fine-tuned models achieve state-of-the-art citation quality, surpassing even advanced proprietary models, including GPT-4o.

**Weaknesses:**

- **Lack of Motivation and Novelty**: In addition to the coarse granularity issue, the authors point out that the chunk-level citation may also result in incomplete sentences. This problem arises because the authors have rigidly divided the context into fixed-length chunks of 128 tokens. A potential solution could be splitting the text based on sentence terminators, such as periods or semicolons. Given that the authors assert that sentence-level citation is superior to chunk-level citation, what motivates the inclusion of chunk-level citation in the CoF pipeline? Additionally, since most methodologies (Evaluation Method / Dataset Construction / Model Training) in this paper are adapted from existing work [1], the novelty is unclear.

- **Obscure and Inconsistent Noun Naming.** There are several confusing noun pairs in the paper. The authors should read through their paper before submission. For instance:
    - *(LongBench-Chat, LongBench-Cite):* In the abstract (line 17), the authors state, "We first introduce LongBench-Cite, ...," whereas in the introduction (line 82), they refer to "We introduce LongBench-Chat, ...". This inconsistency is perplexing, and it is unclear if one of these is meant to reference a published work [1].
    - *(LongCite-9B, LongSFT-9B) and (LongCite-8B, LongSFT-8B):* The naming convention is confusing, making it hard for readers to discern whether these models are synonymous or represent distinct concepts.

[1] Bai, Y., Lv, X., Zhang, J., He, Y., Qi, J., Hou, L., ... & Li, J. (2024). Longalign: A recipe for long context alignment of large language models. arXiv preprint arXiv:2401.18058.

**Questions:**

Could you explain why the chunk size must be fixed at 128 tokens instead of matching the sentence length?

---

> ### Author Response · Authors · 2024-11-19
>
> **About the chunk size**
>
> First, the rigid chunk-divide strategy is directly adopted from previous works such as [1][2]. We just point out their drawbacks, including the coarse granularity and the possibility of producing incomplete evidence sentences.
>
> In addition, we also try strategies that split the text based on sentence terminators as you mentioned, which are just LAC-S, RAC-S, post-LC-S, and post-RC-S introduced in Section 3.2. Their performance is shown in Table 4. The reasons why we still include chunk-level citations in the CoF pipeline are as follows:
>
> 1. From Table 4 we can see that both CoF and post-RC-C achieve better citation F1 than post-RC-S, indicating that using a course-to-fine strategy, i.e, first finding the relevant chunks and then pinpointing evidence sentences from these chunks, is beneficial for improving the performance. Table 9 also shows that SFT with CoF data significantly outperforms that with post-RC-S data.
>
> 2. Another advantage of our course-to-fine strategy is that it can extract continuous evidence sentences such as "[5-7]" from the relevant chunks, resulting in better semantic integrity and coherence, while sentence-level strategies such as post-RC-S often produce too many discontinuous citation numbers (such as "[3][7][15]..."), which makes subsequent learning difficult and is also less user-friendly.
>
> 3. As mentioned in line 323, before extracting the evidence sentences, we expand each relevant chunk by concatenating it with its preceding and succeeding chunks and only retain the complete sentences. This also guarantees that we can extract complete evidence from the relevant chunks.
>
> [1] Enabling large language models to generate text with citations. Gao et al. EMNLP 2023.
>
> [2] Attribute or Abstain: Large Language Models as Long Document Assistants. Buchmann et al. EMNLP 2024
>
> **About the novelty**
>
> In fact, the core parts of our evaluation method (i.e., evaluation for citation quality) and dataset construction (i.e., the citation generation steps in the CoF framework) are totally different from LongAlign, since LongAlign is for long-context alignment and we focus on long-context citation generation instead. We only adopt their evaluation method for correctness and their QA generation method (i.e., the first step in CoF).
>
> We are the first work to study how to construct high-quality SFT data for LQAC task, and our data construction method is novel and carefully designed, significantly outperforming other regular citation generation pipelines. Our improved evaluation method for citation quality also aligns much better than previous methods (see human evaluation in Table 7) in long-context QA scenario. Besides, we propose a new metric correctness ratio, which helps us to find that adding citations via simple in-context learning can harm response correctness while SFT with citation information can improve correctness, which are not mentioned in previous work.
>
> **About the typo**
>
> Thanks for pointing out the typo in line 82. We will correct it in revision.
>
>
> **Thank you for your valuable reviews. Since you may miss some detailed information in our work, we would really appreciate it if you would like to reconsider the score of our paper. Please feel invited to leave more comments in case you have more detailed suggestions and questions.**

---

### Official Review · Reviewer_pvaV · 2024-10-29

**Soundness:** 3
**Presentation:** 4
**Contribution:** 2
**Rating:** 5
**Confidence:** 5

**Summary:**

This paper primarily focuses on the task of Long-context QA with Citation, including the following key points:
1. adoption of sentence-level citation;
2. proposal of an automated data construction method;
3. improved model performance on the LQAC task after training.

**Strengths:**

1. The overall structure is complete, and the writing is well-organized.
2. A comprehensive body of work has been conducted for the LQAC task, including data construction and model training.

**Weaknesses:**

Indeed, as mentioned above, this paper presents a complete body of work around LQAC, with comprehensive content that is self-contained. However, the core issue with this paper lies in its low level of innovation:

1. The proposed LQAC task essentially combines aspects from *short-text QA*, *fine-grained citation*, and *long-context generation with citation*, making the innovation somewhat incremental.
2. I believe that in addition to discussing the general challenges of generation with citation, research on the LQAC task should emphasize the unique characteristics of **long-context scenarios**, such as *citation consistency across contexts*, *citation redundancy*, and *potential contradictions in citations*. This would better highlight the differences between this work and prior research.

**Questions:**

If the authors can clearly and explicitly explain the unique challenges addressed by this work in the long-context scenario, I would consider increasing the score.

---

> ### Author Response · Authors · 2024-11-19
>
> **Regarding the unique challenges we addressed**
>
> Thanks for your valuable review. In fact, our work addresses the following unique and important challenges in the long-context scenario:
>
> 1. **Prolonged user waiting time.**
> Most previous works on fine-grained citation generation rely on relatively complex pipelines that will significantly increase the inference time. For example, [1] breaks down the generation process into three steps: content selection, sentence planning, and sentence generation, and fine-tunes a model for each step. [2] requires the model to first generate all evidence snippets before producing the response. Compared with vanilla QA, these methods more than double the user waiting time, which is unacceptable in long-context scenario. In contrast, our work stays true to first principles and is the first to explore directly employing the long-context LLM to generate in-line citations on the fly (though straightforward, no one ever succeeded before), which basically does not increase the inference time and makes the whole product practicable in real-world usage.
>
> 2. **Compromised response quality.**
> Different from short-context open-domain QA, we first notice that in long-context scenario, one-pass methods (including retrieved-based and long-context-based methods) via in-context learning always hurt the response quality (Table 4). Therefore, we decide to adopt post-hoc strategy in the CoF pipeline, i.e., first obtaining the answer and then adding citations into the answer in subsequent steps, to fully utilize model's long-context capacities. The post-hoc characteristic also allows our pipeline to augment any existing long-context QA datasets with citations. Besides, we also find that SFT with citation information further improves the long-context-based response quality (Table 3), which is also not mentioned in previous works.
>
>
> 3. **Increased learning difficulty.**
> Another challenge of LQAC is the increased learning difficulty caused by the lengthy context and up to thousands of citation indices compared to the short-context scenario. The conventional RAG-based methods (e.g., post-RC-S) for data construction often recall nonadjacent sentences and thus produce too many discontinuous citation numbers (such as "[3][7][15]..."), which would confuse the LLM and make the training difficult. While the course-to-fine strategy in CoF allows extracting adjacent evidence sentences such as "[5-7]" from the relevant chunks, resulting in better semantic integrity and coherence of the cited information. Table 9 shows that using data constructed by our CoF pipeline can achieve significantly better citation quality than conventional RAG-based methods in long-context scenario.
>
> 4. **Inappropriate evaluation method.**
> As shown in Table 6 and 7, we find that previous automatic evaluation method (i.e., ALCE) for short-context citation generation has low alignment with human in long-context scenario because it does not exclude "functional sentences" in the model responses and overlooks partially supporting cases in the calculation of citation precision. In this work, we improve the evaluation method accordingly (Section 2.3.2) and achieve better agreement with human.
>
>
> **Thanks for your valuable feedback again. We would appreciate it if you could recognize our contributions and would like to increase the score of our paper.**
>
> [1] Attribute First, then Generate: Locally-attributable Grounded Text Generation. Slobodkin et al. ACL 2024.
>
> [2] Learning Fine-Grained Grounded Citations for Attributed Large Language Models. Huang et al. Findings of ACL 2024.

---

> > ### Comment · Reviewer_pvaV · 2024-11-26
> >
> > Thank you for the detailed response, but it does not change my assessment of the paper's originality. Therefore, I will maintain the original score.

---

### Official Review · Reviewer_mTi4 · 2024-10-29

**Soundness:** 2
**Presentation:** 3
**Contribution:** 2
**Rating:** 3
**Confidence:** 4

**Summary:**

This paper first evaluates the performance of LLMs in generating sentence-level citations in long-context QA. Then, they distill GLM-4 to construct high-quality fine-grained citation data and fine-tune GLM-4-9B and Llama-3-8B to generate sentence-level citations. Experimental results show that simply SFT can improve citation quality and response correctness.

**Strengths:**

1. The structure of this paper is clear and easy to understand
2. The experimental design (choice of model, dataset, and evaluation criteria) is reasonable.
3. The author noticed that adding citations can harm response correctness, which was not mentioned in previous work
4. Both human evaluation and experiments demonstrate the effectiveness of the trained LongCite-8B and LongCite-9B

**Weaknesses:**

1. Lack of significant novelty: Firstly, the concepts of long-context LLM citation [1] and fine-grained citation [2][3] have already been discussed in previous works. This paper merely combines these two concepts, stitching them together into a single paper. More importantly, the authors even lack discussion and comparison of these important prior works. Secondly, the implementation approach for fine-grained citation heavily overlaps with [2][3], as both rely on distilling advanced LLMs to extract fine-grained spans. The main difference is that this paper extends the context length and distinguishes citation forms by splitting the context into sentences.
2. Unoriginal conclusions: Previous works on attributed text generation have already demonstrated that simply SFT can significantly enhance a model's attribution abilities [2][3][4]. I did not learn anything novel from the authors' conclusions, with the different prerequisite being a long-context.
3. Insufficient benchmarking: The paper only adopts the LAC-S strategy for benchmarking, which is insufficient. It is difficult to deduce whether the model’s poor sentence-level citation quality is more influenced by poor instruction-following or weak evidence-searching ability based on these evaluations.
4. Lack of baseline comparisons: The method in [2] achieves sentence-level citation through distilling Gemini, using different models for content selection and sentence planning. The authors lack a fair comparison with such method.
5. Evaluation of correctness: The authors’ evaluation of correctness includes both accuracy and comprehensiveness, which are actually different dimensions but equally important. These should be separately assessed instead of being unified under correctness. Otherwise, it becomes challenging to discern whether fluctuations in correctness are due to errors or insufficient coverage. The causes and consequences of these two are entirely different. I would prefer referring to correctness as response quality or overall quality.
6. High cost of constructing SFT data: The entire data construction process heavily relies on calling the proprietary model GLM-4, which is not only high-cost but also hard to reproduce.
7. Quality of the question and answer generated by self-instruct: This is crucial for the diversity and quality of CoF data, but the quality cannot be guaranteed. The authors lack specific analysis of the quality of the questions and answers generated in the first step. Will the generated questions focus too much on local information in the background, resulting in insufficient answers?
8. Control over granularity: LongCite-45k contains 128,000 tokens. Assuming an average sentence length of around 20-30 tokens, this would result in over 4,000 sentence indices, which is too fine-grained. I am concerned about its practicality in real-world usage.

I am willing to improve the score if the author can effectively address these concerns.

[1] Attribute or Abstain: Large Language Models as Long Document Assistants. Buchmann et al. EMNLP 2024

[2] Attribute First, then Generate: Locally-attributable Grounded Text Generation. Slobodkin et al. ACL 2024

[3] Learning Fine-Grained Grounded Citations for Attributed Large Language Models. Huang et al. Findings of ACL 2024

[4] Training Language Models to Generate Text with Citations via Fine-grained Rewards. Huang et al. ACL 2024

**Questions:**

1. Typo: Line 82: "LongBench-Chat" should be corrected to "LongBench-Cite".
2. Typo: In Figure 4, lines 802-803, "Example Answer 2" should be corrected to "Example Answer 3".
3. Why are the standards for correctness evaluation inconsistent across different datasets in LongBench-Cite, and why are the scoring ranges different? For example, LongBench-Chat does not consider comprehensiveness, and its score range is 1-10, while GovReport’s score range is 1-5, and others are 1-3. Additionally, why does the evaluation of LongBench-Chat require a few-shot approach, while others are zero-shot?
4. How is the correctness score in Table 3 calculated? Is it the average score across the entire dataset? However, the correctness score range varies across different datasets. Was this unified into a percentage scale here? The authors did not clearly explain this.
5. Why is the performance of GLM-4-9B-chat on the GovReport dataset exceptionally weak in Table 2? Was there any specific analysis conducted?
6. Compared to chunk-level citation, are there any other statistics changes in sentence-level citations, such as the number of citations or the proportion of a statement citing multiple sources

---

> ### Author Response · Authors · 2024-11-21
>
> **W1: Lack of significant novelty & missing related works**
>
> **R:** Thanks for your valuable feedback. We will add these related works in the revision. Nevertheless, we clarify the novelty and difference of our work as follows:
>
> 1. Our work is the first to present a comprehensive body for long-context QA with fine-grained citation, including a new evaluation benchmark, a novel data construction pipeline, and thorough model training experiments to prove the effectiveness. In contrast, [1] is a benchmark rather than a methodology for LQAC. Besides, it focuses on coarse-grained citation and only considers citation recall for free-form QA, overlooking the citation precision. In addition, its measurement method for citation recall, which is adopted from [2], has been shown to have relatively low agreement with human because it overlooks the "functional sentences" in model responses. On the other hand, we improve the automatic evaluation method for citation quality and achieve significantly better alignment with human than [2] in long-context scenario.
>
> 2. Moreover, we address several unique and important challenges of LQAC, making the whole product practicable in real-world usage for the first time:
>
>     (1) **Prolonged user waiting time.** Though [3] and [4] discuss fine-grained citation, their approaches more than double the inference time compared to vanilla QA without citation, which is unacceptable in long-context scenario. [3] breaks down the citation generation into 3 sub-tasks, and [4] requires the model to first output the evidence before generating the response. In contrast, we are the first to explore directly employing long-context LLMs to generate in-line citations on the fly by carefully designing the CoF pipeline for data construction, making the inference time basically the same with vanilla long-context QA.
>
>     (2) **Compromised response quality.** We are the first to notice that one-pass citation generation via in-context learning always hurts the response quality in long-context scenario. Moreover, the content-selection of [3] and the pre-retrieval of [4] will discard most parts of the context, which would further affect the correctness of responses. Therefore, we decide to adopt a post-hoc strategy in the CoF pipeline to fully utilize the model's long-context capacities and guarantee the response quality. The post-hoc characteristic also allows our pipeline to augment any existing long-context QA datasets with citations.
>
>     (3) **Increased learning difficulty.** Another challenge of LQAC is the increased learning difficulty caused by the lengthy context and up to thousands of citation indices compared to the short-context scenario. The conventional RAG-based methods (e.g., post-RC-S and the method of [4]) for data construction often recall nonadjacent sentences and thus produce too many discontinuous citation numbers (such as "[3][7][15]..."), which would confuse the LLM and make the training difficult. While our proposed course-to-fine strategy in CoF allows for the extraction of adjacent evidence sentences such as "[5-7]" from the relevant chunks, resulting in better semantic integrity and coherence of the cited information. Table 9 shows that using data constructed by our CoF pipeline can achieve significantly better citation quality than conventional RAG-based methods in long-context scenario.
>
> **W2: Unoriginal conclusions**
>
> **R:** In fact, we present the following new conclusions that are not mentioned in previous works:
>
> 1. Through LongBench-Cite, we reveal current LLMs’ limited performance on LQAC.
>
> 2. We are the first to notice that one-pass citation generation via in-context learning will hurt the response quality, while SFT with high-quality citation information can further improve the response quality.
>
> 3. We notice that conventional RAG-based methods (e.g., post-RC-S) for data construction lead to too many nonadjacent citation numbers, which will increase the training difficulty in long-context scenerio, and we adopt a course-to-fine strategy in CoF pipeline to address this challenge.
>
> 4. The similar citation length between the trained models and constructed data demonstrates that not only the
> evidence-locating skill but also the citation granularity can be well learned through SFT.
>
> 5. LongCite-8B and LongCite-9B even attain higher citation F1 than the data construction pipeline
> CoF (72.0 and 69.2 v.s. 65.8), implying a potential for iterative self-improvement.
>
> [1] Attribute or Abstain: Large Language Models as Long Document Assistants. Buchmann et al. EMNLP 2024.
>
> [2] Enabling large language models to generate text with citations. Gao et al. EMNLP 2023.
>
> [3] Attribute First, then Generate: Locally-attributable Grounded Text Generation. Slobodkin et al. ACL 2024.
>
> [4] Learning Fine-Grained Grounded Citations for Attributed Large Language Models. Huang et al. Findings of ACL 2024.

---

> ### Author Response · Authors · 2024-11-21
>
> **W3: Insufficient benchmark**
>
> **R:** As we have mentioned in Section 2.4, we select LAC-S strategy as the default setting due to its efficiency, losslessness of context information, and no reliance on additional retrieval systems. It can purely examine the models' LQAC capacities and is also the most ideal strategy in practical usage. In addition, we also list the performance of different strategies in Table 4. By checking the generated responses, we can actually clearly find that: smaller models such Llama-3.1-8B-Instruct and GLM-4-9B-chat suffer more from poor instruction-following ability because they often generate responses that do not conform to the prescribed format, while larger models such as Llama-3.1-70B-Instruct and Mistral-Large-Instruct are influenced more by weak evidence-searching ability.
>
> **W4: Lack of baseline comparison**
>
> **R:** We evaluate the best few-shot method in [3], i.e., Attr. First CoT, on LongBench-Chat, using their open-sourced code and taking GLM-4 as the backbone LLM.
> |  | Citation recall | Citation precision | Citation F1 | Correctness | Correct. Ratio |
> |---|:---:|:---:|:---:|:---:|:---:|
> | CoF | 61.3 | 81.8 | 66.1 | 79.8 | 100% |
> | Attr. First CoT | 40.9 | 31.8 | 34.3 | 33.0 | 41.4% |
>
> We find that when the context becomes longer, the content selection step of [3] often fails to locate the most relevant information, especially for some complex queries. This leads to bad response quality as well as low citation quality.
>
> **W5: Evalution of correctness**
>
> **R:** Thanks for your advice. Since our evaluation of correctness follows the tradition of long-context benchmarks such as [5][6][7], which typically score the model response with the golden answer, we will change the definition of correct into "Whether the response is accurate and consistent with the groundtruth".
>
> **W6: High cost of constructing SFT data**
>
> **R:** Considering that our goal is to achieve better performance than the most advanced proprietary model, the cost of data construction is acceptable, especially compared with human annotation. In addition, we have open-sourced our code, models, and LongCite-45k dataset for reproducing and further research.
>
> **W7: Quality of the question and answer generated by self-instruct**
>
> **R:** We directly employ the QA generation method proposed by [5], which has been proven high-quality in their paper. Our experiments also show the SFT models using the self-instructed data achieve comparable or even better long-context QA performance (i.e., $\text{C}_\text{LQA}$) than their officially post-trained counterparts. The good performance on multi-doc QA also indicates that the generated questions do not focus too much on local information.
>
> **W8: Control over granularity**
>
> **R:** In real-world usage, when the user clicks on the citation number, the corresponding text will be highlighted in the original context, without the need to find the evidence by the user himself/herself. The human evaluation in Table 6 and our online test have shown that our system has good practicality.
>
> **Q1 & Q2: Typos**
>
> **R:** Thanks for pointing out the typos. We will correct them in the revision.
>
> **Q3 & Q4: Evalution for Correctness**
>
> **R**: We just directly adopt the official evaluation methods and prompts for correctness from [5]. Since LongBench-Chat is a harder and smaller benchmark with long-form groundtruth, the authors of [5] set a score range of 1-10 for better discriminability and **annotated 3 scoring examples for each test instance** as the pivots for better alignment with humans. While the tasks in LongBench are relatively easier(the context is shorter and the groundtruths of QA tasks are short phrases), so zero-shot approach and smaller score ranges are enough. In Table 3, the correctness scores are unified into a percentage scale, and the average score is the average of the five datasets.
>
> **Q5: Performance of GLM-4-9B on GovReport**
>
> **R:** Thanks for pointing out the exception. By checking the generated response, we find that for 31 out of 200 test instances of GovReport, LongCite-9B directly outputs the summary without generating citations. A feasible fix method is to force the model to first generate the special token "\<statement\>" so that it will generate citations for each instance. This leads to 80.7 citation F1 on GovReport.
>
> [3] Attribute First, then Generate: Locally-attributable Grounded Text Generation. Slobodkin et al. ACL 2024.
>
> [5] Longalign: A recipe for long context alignment of large language models. Bai et al. Findings of EMNLP 2024.
>
> [6] LongBench: A Bilingual, Multitask Benchmark for Long Context Understanding. Bai et al. ACL 2024.
>
> [7] ∞Bench: Extending Long Context Evaluation Beyond 100K Tokens. Zhang et al. ACL 2024.

---

> > ### Author Response · Authors · 2024-11-21
> >
> > **Q6: Statistic changes**
> >
> > **R:** Here are some statistis changes, using GLM-4-9B as the base model for SFT:
> > |  | #citation per response | #citation  per statement | proportion of statements citing multiple sources |
> > |---|:---:|:---:|:---:|
> > | chunk-level | 7.62 | 1.05 | 29.3% |
> > | setence-level | 6.06 | 0.76 | 10.4% |

---

> > > ### Author Response · Authors · 2024-11-21
> > >
> > > **Thanks for your valuable feedback again. If our response addresses your concern, we would appreciate it if you would like to increase the score. Please feel invited to leave more comments in case you have more detailed suggestions and questions**

---

> > > > ### Comment · Reviewer_mTi4 · 2024-11-30
> > > > **Official Comment by Reviewer mTi4**
> > > >
> > > > Thank you for the detailed reply. Although the authors' responses address some of my concerns, I still have doubts about the novelty of this work and the originality of the conclusion (refer to weaknesses 1 & 2). Despite the authors highlighting some of their contributions, I believe the core of the paper appears too incremental compared to existing research [1][3][4] (as cited in the authors' response). Based on the current response, I maintain my original score.

---

### Official Review · Reviewer_GQxi · 2024-11-04

**Soundness:** 3
**Presentation:** 2
**Contribution:** 2
**Rating:** 6
**Confidence:** 4

**Summary:**

This paper introduces LongBench-Cite, a Long-Context QA benchmark that requires models to add citations in their responses. Different from other similar works, this work evaluates citations at the sentence level, rather than at the chunk level. That is, models in this task are supposed to cite specific sentences in context that support their responses rather than documents or large chunks of text. The authors also construct LongCite-45k, which is a supervised fine-tuning dataset for the task studied, and use this dataset to train two models. Their 8B and 9B parameter, fine-tuned models outperforms GPT-4o. The results of their experimentation show that existing models are not proficient in sentence level citation. Moreover, requiring sentence level citations degrades model performance. Finally, fine-tuning improves model capability in QA and sentence level citations a reasonable amount.

**Strengths:**

**Interesting Task.** Studying fine-grained citations (at the sentence-level) is interesting. Efforts in developing models with the capability to provide fine-grained citations is useful. In particular, this could help with evaluation of the veracity of model outputs

**Results.** The results show that the fine-tuned models generally outperform proprietary models in long-context QA with citations, which is a nice result. Furthermore, results seem to imply that proprietary models may have been adequate at chunk-level citations but struggle more with sentence level citation. Finally, the "correctness ratio" measurement, which compares how models perform with and without having to cite, is interesting and clearly shows shows that making the models output citations alongside correct answers seems to degrade model performance on QA.

**Complete Work.** The paper presents a relatively complete exploration of an idea: a new variant of QA with citations is proposed; an evaluation metric for the task is proposed; existing models are evaluated on the task; a dataset is constructed for fine-tuning, and used to train new models for the task; those models perform well; analysis is presented.

**Human evaluation** It’s great to have human evaluation. I liked the analysis of LongCite-9B vs LongSFT-9B; I think it could be fleshed out more. The manual evaluation of citation recall/precision strengthens the paper, although I think the correlation is not entirely convincing since--if I understand correctly--it only includes 3 data points (GLM-4, LongCite-8B and -9B). The presentation could perhaps be strengthened; it took me some time to correctly interpret the correlation from the table. The Cohen’s Kappa result is more convincing, although I wasn’t sure how it was computed (questions below).

**Weaknesses:**

**Clarity and Missing Details.**  The writing is a bit hard to follow at times, for example in the description of the construction of the dataset (see questions below). Another area that was hard to follow were the pipeline details. Specifically, what retrieval method is used to retrieve chunks in the chunk-level citation generation (I know it's Zhipu, but could more details be provided)? How were the hyperparameters chosen (l, k, & n)? In pipeline validation, how are the evaluations carried out (done manually? Or automatically)?

**Over-reliance on LLM Evaluation.** Either GPT-4o or GLM-4  is used for: 1) evaluating correctness of QA answers, 2) evaluating correctness of citation (precision and recall), 3) filtering out starting, transition, summary and reasoning sentences, 4) all steps of the CoF pipeline for dataset construction. Your human evaluation showing correlation between GPT-4o evaluations and human evaluations is critical to dispelling some concern over possible systemic errors from GPT-4o corrupting your results, but the concern isn’t totally alleviated. For example, it seems like one way to get a high score is to emit many sentences that GPT-4o would deem “starting, transition, summary or reasoning” since these need no citations according to the evaluation scheme. I think a discussion of this reliance and where things could go wrong should be included in a limitation section, or additional measurements and/or human evaluations should be conducted to address potential issues.

**Missing Discussion of Sensitivity to Prompt.** Since the proprietary models are only prompted, and the task itself has a somewhat atypical format (e.g., with each sentence tagged with an index) it would be good to discuss whether any tuning of the prompts was conducted (especially for the proprietary models) and/or any observations related to the sensitivity of the models to the prompt. This would help alleviate concern that the proprietary may have performed better if only their prompts were tuned better.

**Questions:**

- Construction of LongBench-Cite: Is LongBench-Cite just a concatenation of existing datasets? Is there any additional processing to create LongBench-Cite examples (and if so what is it)? If it is just a concatenation, then what is the advantage of combining it into one dataset vs. evaluating on them each separately?
- Can you explain precisely what is being compared in the Cohen’s Kappa calculation? Is it correctness at the response level, or citation quality at the sentence level or something else?
- Can we just use CoF to do the LQAC task? What would be the disadvantage?
- Not a question: When you list multiple Figures, often only one is hyperlinked in the text; all should be hyperlinked
- If you use GPT-4o for all of the evaluation steps, is it fair to compare GPT-4o (and have it evaluate itself)?
- Did you notice any “lost-in-the-middle” type phenomenon (https://arxiv.org/abs/2307.03172)?
- How much do you think the format of the citations you are asking for affects the sentence-level citations results of proprietary models? In other words, do you think there is a way to reformulate the task (prompt) such that those models would perform much better? Did you try many different prompts?
- Is it possible/necessary for your models to cite non-contiguous sentences, e.g., [1-2;5] ?

---

> ### Author Response · Authors · 2024-11-21
>
> Thanks for your valuable feedback.
>
> **W1: missing details**
>
> The retrieval method is simple dense retrieval, using Zhipu embedding-2 as the embedding model. Specifically, for each sentence in the model response, we retrieve the top-$l_\text{max}$ chunks whose embeddings are most similar to the sentence's embedding. The hyperparameter $l_\text{max}$  and $k$ ($n_\text{sent}$ is the number of sentences in the answer rather than a hyperparameter) are chosen based on several samples from the training set so that the retrieval recall of evidence chunks surpasses 90%.
>
> In pipeline validation, the evaluations are carried out automatically, using the evaluation method described in Section 2.3.
>
>  **W2: Over-reliance on LLM evaluation**
>
> Thanks for your advice. As mentioned in the paper, we rely on LLM evaluation because the conventional evaluation methods are no longer suitable for the LQAC scenario and have low agreement with human. In addition, a possible method to defend the hack you mentioned is to ignore these "functional sentences" when calculating the citation recall.
>
>  **W3 & Q7: Regarding the prompt format**
>
> In fact, we have tuned the prompt so that the models can achieve better performance.  There are three key points for the prompt tuning:
>
> 1. As shown in Figure 10, we repeat the instruction at the end of the prompt, after the context and question presented. This reminder of instruction is helpful for the models to follow the instruction better, especially when the context is too long.
>
> 2. We require the models to generate citations in the format of spans such as "[3-5][8-9]" instead of conventional separated indexes such as [3][4], since we found that the latter often leads to non-stopping enumeration (e.g., [3][4][5][6][7]...).
>
> 3. The demonstration contains citations from each part of the context so that the models would not focus too much on local information.
>
> **Q1: Construction of LongBench-Cite**
>
> Yes, LongBench-Cite is a concatenation of existing datasets from LongBench-Chat and LongBench. We combine them for unified and convenient evaluation and benchmarking. A difference is that we require the models to generate long-form responses for all instances,  while the original LongBench asks the models to only output the answer phrase for the QA tasks, which is not suitable for citation generation.
>
> **Q2: Regarding the Cohen's Kappa**
>
> As shown in Table 7, the citation recall/precision at the sentence/citation level measured by GPT-4o and human are compared in the Cohen's Kappa calculation.
>
> **Q3: Using CoF to do the LQAC task**
>
> Yes, CoF can be directly used for the LQAC task. However, the main disadvantage is that the complex pipeline will significantly prolong user waiting time, while the fine-tuned models can generate the in-line citations on the fly, which basically does not increase the inference time compared to vanilla long-context QA.
>
> **Q4: Regarding the GPT-4o evaluation for itself**
>
> In the evaluation for correctness, there is indeed a risk that GPT-4o will assign higher scores for itself, as found by previous works. However, since we focus more on the correct ratio, which is a relative ratio, the influence will be alleviated. In addition, because the evaluation of citation recall and precision is similar to sentence-level NLI tasks, GPT-4o would not present obvious bias.
>
> **Q6: Regarding the "lost-in-the-middle" phenomenon**
>
> Yes, we notice some "lost-in-the-middle" phenomenon for vanilla LongSFT models. Though [1] has shown that using self-instruct QA data can achieve all-green performance on the simple "needle-in-a-haytask" test, we found that when faced with a query that requires a global view, LongSFT models tend to use more information from the head part of the context and only roughly utilize or even ignore the rest of the context (Section 4.2.1). In contrast, the generated citation numbers allow LongCite models to be aware that current response content has covered which parts of the context, so that they can utilize different parts of context more uniformly, resulting in a more comprehensive response.
>
> **Q8: Citing non-contiguous sentences**
>
> Yes, our models support such citations such as [1-2][5-5], as shown in Table 10.
>
> [1] Longalign: A recipe for long context alignment of large language models. Bai et al. Findings of EMNLP 2024.

---

> > ### Comment · Reviewer_GQxi · 2024-11-25
> > **Thank you for the response**
> >
> > Thank you for responding to my concerns. I suggest that you include the various items you raised in a future revision of your paper.

---

### Author Response · Authors · 2024-11-25

Dear Reviewers:

Thank you again for your helpful reviews. As the end of the discussion period draws near, we would like to ensure that we have adequately addressed all your concerns. If you have any further feedback, please do not hesitate to let us know.

---

### Note · Authors · 2024-12-14

I have read and agree with the venue's withdrawal policy on behalf of myself and my co-authors.